# A New Hypothesis Describing the Pathogenesis of Oral Mucosal Injury Associated with the Mammalian Target of Rapamycin (mTOR) Inhibitors

**DOI:** 10.3390/cancers16010068

**Published:** 2023-12-22

**Authors:** Stephen T. Sonis, Alessandro Villa

**Affiliations:** 1Divisions of Oral Medicine and Dentistry, Brigham and Women’s Hospital and the Dana-Farber Cancer Institute, Boston, MA 02115, USA; 2Department of Oral Medicine, Infection and Immunity, Harvard School of Dental Medicine, Boston, MA 02114, USA; 3Biomodels, LLC, Waltham, MA 02451, USA; 4Oral Medicine, Oral Oncology and Dentistry, Miami Cancer Institute, Miami, FL 33176, USA; 5Herbert Wertheim College of Medicine, Florida International University, Miami, FL 33199, USA

**Keywords:** mTOR inhibitor, rapamycin, everolimus, oral ulceration, aphthous, cancer

## Abstract

**Simple Summary:**

The mammalian/mechanistic target of the rapamycin (mTOR) pathway is made up of many components that have far-reaching biological consequences, such as cell division, growth, and metabolism. The results of experiments performed years ago showed that inhibitors of the mTOR pathway have a negative effect on tumor cells, which sparked interest in the development of a class of drugs called mTOR inhibitors as anti-cancer therapies. mTOR inhibitors are now part of oncologists’ armamentarium for various types of cancers, but patients who use these drugs are at a high risk of developing painful mouth sores. In this paper, we provide a hypothesis as to why these sores develop, how their development compares to other common mouth sores such as canker sores, why other parts of the body are not affected, and how treatment may work.

**Abstract:**

It has been 24 years since rapamycin (sirolimus) was approved to mitigate solid organ transplant rejection and 16 years since mTOR (mammalian/mechanistic target of rapamycin) inhibitors reached patients as a cancer therapy. While the clinical benefits of mTOR inhibitors (mTORi) are robust, so too are their toxicities. Among the most common issues is the development of ulcers of the oral mucosa (mTOR-inhibitor associated stomatitis; mIAS). These lesions are distinct from those of other anti-cancer agents, occur with regularity, and impact patient outcomes. mIAS’ pathogenesis has been the subject of speculation, and its similar presentation to recurrent aphthous stomatitis (RAS) has led to the hypothesis that it might serve as a surrogate to better understand RAS. Based on a review of the literature, the current manuscript provides a hypothesis regarding the mechanisms by which mTORis uniquely initiate mucosal injury and an explanation for the observation that steroids (also an immunosuppressive) are effective in its treatment through a non-immunologic mechanism. Unexplained unique features of mIAS are discussed in this review in the context of future investigation.

## 1. Introduction

The mammalian target of the rapamycin (mTOR) pathway has been said to control virtually every cellular function [1]. In truth, the mTOR pathway is complex, with multiple functionalities, and has become a clinically important pharmacological target. The development of mTOR pathway inhibitors has resulted in the creation of several multigenerational drugs (mTOR inhibitors; mTORi) that have toxicities that commonly impact the oral mucosa. The pathogenesis of mTOR inhibitor-associated stomatitis (mIAS) has not been fully investigated. Indeed, the seemingly mechanistic biological paradoxes involved in these lesions are compelling. Equally perplexing is the observed effectiveness of corticosteroids in managing lesions for which the etiology is driven by drugs (mTOR inhibitors) with known immunosuppressive activity. While the clinical presentation of mIAS and the lesions associated with conventional cytotoxic cancer therapy (oral mucositis) is different, certain aspects of lesion induction are similar. Furthermore, the phenotypic similarities between mIAS and recurrent aphthous stomatitis beg the question of pathoetiologic similarities.

Among the questions of interest are the following:Do mTORi cause direct cell death and, if yes, by what mechanism?If mIAS is initiated by mTORi-mediated direct cell death, why is the mucosal lesion presentation different from conventional cytotoxic therapy?Is there a relationship between the level of mTORi immunosuppression and mIAS?What drives the anatomic site predilection for mIAS and why, given the systemic nature and pharmacodynamics of treatment, and are oral lesions not widespread?Is there a relationship between the dermatologic toxicities of mTORi and mIAS?Why is the oral mucosal predisposed to mTORi-mediated injury compared to another stratified squamous epithelium, i.e., the vaginal mucosa?Is mIAS primarily mediated by the mTORC1 or mTORC2 pathways? Or does it require both?How closely does mIAS resemble aphthous pathobiologically?Is autophagy vs. apoptosis a component of mIAS pathogenesis?Is the efficacy of corticosteroids as an intervention for mIAS independent of their immunosuppressive activity?

The objective of this manuscript was to leverage data derived from an extensive review of the current literature to develop possible hypotheses relating to the unique pathogenesis of mIAS and simultaneously identify biological conflicts to provoke additional study.

## 2. Background

The mTOR story began on Rapa Nui, a small island in the eastern Pacific Ocean most associated with its archeology and the name Easter Island when, in 1964, a group of Canadian scientists collected soil samples in an attempt to find naturally occurring antibiotics. In 1972, Dr. Suren Segal at Ayerst Laboratory’s Montreal laboratory, isolated *Streptomyces hygroscopicus*, which was similar to a macrolide lactone, named it rapamycin based on its site of discovery, and began its development as an anti-fungal [2]. However, the additional characterization of the isolate revealed that it had immunosuppressive activity. While this finding quashed its anti-fungal utility, it opened rapamycin’s use in the management of solid organ transplant rejection, activity that led to it being approved for this purpose in 1999 by the U.S. Food and Drug Administration [3,4].

In the meantime, although Ayerst lost interest in the rapamycin (also known as sirolimus) program, Dr. Segal remained enthusiastic about its potential, temporarily storing samples in his home freezer; this was until 1972, when following Ayerst’s merger with American Home Products, he was able to resume his investigation, in which he noted that rapamycin, among other activities, has broad anti-tumor potential. This led to a subsequent collaboration with the Developmental Therapeutics Program of the U.S. National Cancer Institute (NCI), where screening against a broad range of tumor lines demonstrated its utility as an inhibitor of tumor proliferation [5]. As a consequence, NCI concluded that rapamycin should advance as a priority drug.

Subsequent studies to identify and characterize rapamycin’s anti-proliferative effect ascribed the drug’s effect to its binding to target proteins, which were designated Target of Rapamycin (TOR) [2]. Studies in the 1990’s identified two genes associated with TOR (TOR1 and TOR2) activity. Further investigations led to the molecule’s purification and characterization in mammals, leading to its designation as mTOR (mammalian target of rapamycin) and the recognition that it is an atypical serine/threonine protein kinase belonging to the phosphoinositide 3-kinase (PI3K)-related family [1]. Critically, from the standpoint of activity and its impact on toxicities, mTOR reacts with different proteins to form two complexes (mTORC1 and mTORC2), which differ in functionality and mechanisms [6] but have the capacity for crosstalk. Details describing the composition and specific differences between the complexes are the subject of many excellent reviews on the topic and beyond the scope of this paper. Nonetheless, from the standpoint of stomatitis pathogenesis, the recognition that there are differences between the two complexes in their susceptibility to specific mTOR inhibitors is important. Essentially, the mTORC1 complex is associated with cell proliferation and growth, protein synthesis, lipogenesis, autophagy, and energy metabolism. Its overactivation is associated with tumor development and progression. mTORC2 is thought to be responsible for cytoskeletal organization, lipolysis, and insulin sensitivity [7].

The finding that numerous tumors manifest abnormal mTOR expression led to its potential inhibition as a novel anti-tumor strategy. First rapamycin (sirolimus) and then a series of “rapalogs” (rapamycin and its analogs) were developed and enabled for oncology. These included everolimus, temsirolimus, and ridaforolimus, of which the first two are approved by the FDA for the treatment of specific malignancies, including breast and renal cell cancers. While it was generally agreed that the primary target of these mTOR inhibitors was the mTORC1 complex, for which rapamycin was originally thought to be specific, it now appears that these agents also impact mTORC2 [8]. This finding was especially important from the standpoint of mTORi-associated toxicities in that side effects were generally ascribed to mTORC2.

## 3. mTOR Inhibitor Stomatitis Background

### 3.1. Clinical Presentation, Incidence and Risk

As recalled above, the first approved clinical purpose for an mTORi (sirolimus) was based on its observed immunosuppressive activity in the prevention of solid organ transplant rejection. Shortly thereafter, an association between sirolimus use and oral ulceration was reported [9]. While the severity of these lesions was dramatic, their etiology was not definitive, as some were ascribed as being herpetic in nature, although it was confirmed that cultures were absent. The authors contemplated additional mechanistic causes, including a sirolimus-induced reduction in cell proliferation, its effect on growth factors, or possibly a form of vasculitis.

It was mTORi’s onco-therapeutic potential that drew the attention of further investigations regarding its association with stomatotoxic side effects. Using safety data obtained from two Phase 1b clinical trials in which deforolimus was studied in solid tumor patients, Sonis and his colleagues first characterized the oral lesions associated with mTORi in cancer patients as being distinct from the lesions of mucositis observed in patients receiving cytotoxic cancer therapy [10]. Mucosal ulcers, identical in appearance to those associated with recurrent aphthous stomatitis (RAS; described as ovoid ulcer with an erythematous halo), were observed in 66% of the 78 patients studied (Figure 1). In general, ulcers occurred within 5 days of deforolimus administration and, in contrast to conventional mucositis, were associated with nonspecific rashes or acneiform dermatitis, but not gastrointestinal adverse events. To differentiate these lesions from those associated with chemotherapy or radiotherapy, the term mTOR inhibitor-associated stomatitis (mIAS) was applied.

While the clinical presentation of aphthous and mIAS is similar, their course and pathoetiology are not. Both lesions are characterized by well-demarcated mucosal ulcerations surrounded by an erythematous halo. The lesion (aphthous) on the left is a few days earlier in progression than the mIAS lesion.

mIAS has been further described and cataloged by many investigators for a range of mTORis [11,12,13,14]. Interestingly, the reported incidence of all grade mIAS is relatively similar for the rapalogs everolimus, temsirolimus and ridaforolimus, ranging from 44% to 60.8%, which is markedly higher than that reported for sirolimus (Table 1). Severe mIAS is relatively uncommon for the rapalogs (range 3–9%) and minimal for sirolimus. Aphthous-like ulcers were reported in only 2.6% (1 of 39) of cardiac transplant patients receiving immunosuppressive therapy with sirolimus vs. 15.6% (5 of 32) treated with everolimus [15]. When assessing the incidence data for mIAS, it should be noted that most reports are derived from the adverse events documented in clinical trials, which are often underreported. This point is highlighted by the disconnect between the reported incidence of severe mIAS (maximum 9%) and it being cited as the primary reason for dose de-escalation (27.3%; 48/176 patients in clinical trials) or the discontinuation of mTOR therapy (13.1%; 13/99 patients in clinical trials) [11]. Additionally, as was the case with the initial reports of oral ulcers associated with rapamycin use, errors in diagnosis and the lack of an established scoring system are confounding. Nonetheless, mIAS occurs with a frequency significant enough to be clinically impactful. The differences in stomatotoxicity between the rapalogs and sirolimus are important as we consider the mechanisms by which it occurs.

As noted above, the effects of mTOR are broad and can involve metabolism. It is not surprising, therefore, that hyperglycemia is a common side effect of everolimus [16]. Could this impact mIAS risk? In a meta-analysis of a total of 1455 patients, Rugo and her colleagues examined potential mIAS risk factors and noted that patients with *no* prior history of diabetes had both a higher rate of mIAS (68%) than those with a history (59%) and developed mIAS more quickly [17]. It is also noteworthy that everolimus can be associated with new-onset diabetes, typically in the first six weeks of dosing, but often later. The timing of this event is later than that noted for mIAS. Given the negative relationship between mIAS risk and diabetes history and the differences in the trajectory of mIAS and everolimus-associated new-onset diabetes, it would seem the two are not pathoetiologically related.

### 3.2. mIAS Trajectory and Pharmacokinetics

Critical to trying to define the mechanism by which mTORis are stomatotoxic are observations describing the trajectory and pharmacokinetics associated with mIAS. It appears that the onset of most cases of mIAS occurs shortly after the administration of mTORi, typically within the first cycle of treatment (cycle typically 28 days), and often in the first week [11,16]. In a small but granular study of 17 patients treated with everolimus or ridaforolimus, de Oliveira et al. reported a median time to onset of 10 days after the start of therapy (range: 4–25 days) [14]. In a comprehensive review of the subject, Martins et al. reported that the mIAS frequency was associated with the dose intensity and route of administration [11]. Similar findings were noted by Ferte et al., who also confirmed a dose response (everolimus) relative to incidence, time to onset and duration of mIAS, as well as the impact of antecedent or concomitant chemotherapy on increasing mIAS risk [18]. Importantly, while many cases are short lived, extended mIAS is not uncommon.

### 3.3. mIAS Pathogenesis

The far-reaching biological and functional effects of mTOR and the mTOR pathway have been well described and are still evolving. Understanding the complexity of the pathway and how its inhibition provokes mucosal injury has yet to be determined. Furthermore, the clinical presentation of mIAS has stimulated hypotheses that suggest the possibility of its pathobiological resemblance as a surrogate for RAS and its related conditions (Behcet’s disease). The rapidity with which mIAS may develop in some cases compared to its slow evolution in others might provide some insight into how it evolves. Furthermore, the nonspecific phenotype not only resembles aphthous stomatitis, but other forms of mucosal ulcerations, particularly those associated with epithelial injury.

### 3.4. Initiation of mIAS

For the sake of defining the pathobiology of mIAS, it might be useful to divide its course into three stages: induction, the active ulcerative stage, and resolution. It would seem that mIAS’ induction is the consequence of direct cell injury and the stage at which mIAS is most different from either oral mucositis (OM) or RAS. Using an organotypic model of human mucosa, which contained both epithelial and submucosal elements (Langerhans cells), Sonis and his colleagues observed that everolimus elicited changes consistent with epithelial injury in response in a dose-dependent way. Following a 24 h incubation, epithelial disorganization, pre- and early apoptotic changes, increased TUNEL (terminal deoxynucleotidyl transferase dUTP nick end labeling)-positive staining (an indication of apoptosis), the reduced proliferation of cell nuclear antigen (PCNA)-positive cells (a measure of proliferation) and increases in IL-6, but not other pro-inflammatory cytokines (IL-1β, IFNγ, IL-α, IL-8, TNFα), were significantly impacted [19] (Figure 2 and Figure 3). The conditions under which the experiment was performed precluded the presence of bacteria. The study results suggested that mIAS initiation was the consequence of everolimus-mediated direct cell injury (Figure 1), was dose-dependent, and was independent of the microbiome. The necessity of supporting cells in the lamina propria was not determined.

Because of the phenotypic similarities between aphthous and mIAS, it has been suggested that the two are mechanistically similar [20]. Likewise, the dissimilarity in the presentation of OM and mIAS has implied differences in their pathogenesis [10]. It appears that the mechanism(s) by which mTORis initiate mucosal injury is thus distinctive from those noted for both RAS and OM. While the activation of the innate immune response has been implicated in both RAS and OM [21,22], this is not the case with MIAS. Not only does mTOR inhibition suppress TLR3 activation, its effect on TLR3 results in the regulation of inflammatory processes, as shown in human keratinocytes [23].

So how does the process leading to mIAS occur? Typically, the ulceration of the oral epithelium is a direct or indirect consequence of one of three processes: necrosis, apoptosis, or autophagy [24]. While apoptosis and autophagy are programmed, necrosis is typically initiated by an exogenous source such as a chemical burn. Clinically, necrotic lesions of the oral mucosa are not dissimilar to those of aphthous or mIAS, although they are more random in distribution and less defined. Apoptotic cell death underlies the pathogenesis of many types of oral ulcerations, including those associated with RAS and OM associated with cytotoxic cancer therapies.

But while apoptosis is common to all three conditions, its unique initiation and contribution to ulcer progression is different. While abnormal apoptosis has been suggested to be associated with a damaging cascade associated with the innate immune response for aphthous, the contribution of apoptosis to mIAS is different [21]. And mTORI’s anti-proliferative activity is even more impactful when it is overlaid on the death of renewing epithelial cells.

B-cell lymphoma-2 (Bcl-2) is a family of proteins that regulate cell death by blocking apoptosis [25]. Thus, its expression is viewed as a pro-survival protein. Both rapamycin and everolimus have been noted to not only inhibit Bcl-2 expression, but also to induce apoptosis; in addition, everolimus’ ability to significantly decrease pro-survival proteins has been reported to occur even with low concentrations of the drug [26]. This is most likely associated with the inhibition of the mTORC2 complex given its role in cell survival and the observation that oral toxicities of rapamycin, incorrectly long thought to be mTORC1-specific, generally are not seen until late in dosing. Thus, it seems likely that everolimus-associated apoptosis may be a component of mIAS initiation. The observation that Trefoil Factor 1 (TFF1) effectively prevented apoptosis supports this hypothesis [27].

However, at least two additional mechanisms may play a role in the first phase of mIAS development. Macrophages are normal residents of the oral mucosa’s lamina propria, playing a critical role in the local immune response as antigen-presenting cells [28] and in effecting the local inflammatory response. Martinet et al. [29] reported that everolimus increased the phosphorylation of p38 MAPK, which resulted in the secretion of pro-inflammatory cytokines including IL-6 and, in agreement with other investigators, noted the everolimus was an effective inducer of autophagy.

Under normal conditions, autophagy is a catabolic process in which cytoplastic material is transferred to the lysosome for degradation and recycling [30] in response to cellular stress. Curiously, while considered to be a cell survival pathway for stressed cells, autophagy may also lead to apoptosis [31]. The relationship between autophagy and apoptosis may be independent, occur in parallel, or be synergistic, but may also be a relationship in which autophagy initiates apoptosis. The mTOR pathway plays a critical role in autophagy [32,33] and, from the standpoint of mIAS, the observation that everolimus elicits autophagy seems relevant [31], especially in the context of its potential role in eliciting apoptosis.

At least two other factors suggest both a dissimilarity and a likeness between RAS pathogenesis and that of mIAS. It has been proposed that the ulcers of RAS are a consequence of massive transepithelial apoptosis in which secondary necrosis leads to DAMP (damage-associated molecular patterns) signaling and the initiation of the innate immune response, as noted above. Consequently, Al-Samadi et al. characterize RAS development as being a “top down” phenomenon [21]. This would seem to contrast with the potential initiating/promoting mechanisms of mIAS, which would support a basal stem cell target. In both instances, massive apoptotic changes and a lack of renewal would lead to ulceration. In contrast, caspase-3 expression in all epithelial layers was deemed to be indicative of the “full thickness” epithelial apoptosis described for RAS. In concordance with this finding are the results of studies in which increases in caspase-3 were noted following everolimus [34,35]. While it would be theoretically possible that, as has been described for RAS, necrotic epithelial cells could amplify pro-inflammatory pathways via the innate immune system, the mTORi attenuation of this pathway and the dosing schedule associated with mTORi use seem to make this unlikely.

### 3.5. Ulceration and Resolution and Rationale for Steroids

While there do not seem to be descriptions of the histological changes associated with mIAS ulcers, both animal data (Sonis et al. Unpublished) and the clinical phenotype indicate that the tissue changes are consistent with the well-described non-specific features associated with traumatic or aphthous lesions. Lesions in both conditions are painful but resolve spontaneously in two weeks or less [36]. Since everolimus is administered with a daily dosing schedule, mIAS’ time to onset being typically soon after the initiation of treatment, but generally with the first two months, suggests a rapid biological effect. The temporary discontinuation of the drug is associated with resolution and, in a finding which contradicts logic, the incidence of mIAS appears to decrease in specific patients with second courses of treatment. This finding is in sharp contrast to the increased risk of mucosal toxicity in patients receiving multiple cycles of cytotoxic chemotherapy. Clearly, additional studies will be needed to more fully understand this observation.

Topical steroids have been a mainstay in the treatment of inflammatory oral diseases such as aphthous and lichen planus. They are not effective in the management of either radiation- or chemotherapy-induced mucositis but have been shown to be of benefit in the case of mIAS, as was first demonstrated in the Phase 3 clinical trial for Ridaforolimus (NCT00538239). The rationale for topical steroid use was based on their efficacy in controlling inflammation. However, since steroids also have marked immunosuppressive activity, their use in managing ulcerations induced by another immunosuppressive agent has always seemed to be a clinical oxymoron. In fact, steroids’ anti-inflammatory activity is somewhat distinct from their immunosuppressive action, and it is their anti-inflammatory activity that is attractive for mIAS [37]. In particular, steroids (dexamethasone) effectively modulate the mTOR signaling pathway to interfere with the release of proinflammatory cytokines. This conclusion was reached when the effects of clobetasol were observed in response to everolimus [29,38]. Consequently, it is not surprising that topical steroids (dexamethasone) have been successful as an intervention for mIAS [39], with two caveats. Based on mIAS pathogenesis and clinical observations, topical steroid therapy is most appropriate as a treatment, not a prophylaxis. Second, a topical steroid formulation is preferred over a systemic route of administration.

## 4. Conclusions

While the enthusiasm for mTORis in the oncology world is threatened by newer approaches such as immunotherapy and antibody–drug conjugates, Afinitor (everolimus) sales are expected to be USD 2.8 billion in 2023 and reach USD 3.6 billion by 2030 [40]. Therefore, mIAS is likely to be a component in the management of cancer patients, particularly those being treated for renal cell and breast cancers, for the foreseeable future. However, while we have made some headway in better understanding its pathobiology and its associated clinical implications, a few questions remain.

A perplexing question about mIAS has been its seeming predilection for the oral mucosa, while a similar condition spares other seemingly similar mucosal tissues such as the vaginal mucosa. One explanation might be that the macrophage and possibly the dendritic cell populations are distinct between the two forms of mucosa [39,41]. While no head-to-head comparisons have been performed, studies of non-oral areas of the gastrointestinal tract suggest differences. Likewise, the numbers and characteristics of dendritic cells might also vary [42,43]. An obvious difference between the two sites is the microbiome, but the lack of substantive evidence to suggest its role in the development or course of mIAS detracts from its significance. And, in fact, the topical administration of a mouthrinse containing doxycycline failed to markedly alter the course of mIAS [44]. Nonetheless, additional studies to better understand the role of the microbiome in effecting the course of mIAS may be justified, especially in patients in whom mTORi is administered over a prolonged period, as animal data indicate that chronic mTORC2 suppression impacts the gut microflora [45]. Conversely, a better understanding of the reasons why mIAS is often a coincident finding with mTORi-related dermatologic side effects would be informative.

And finally, given the frequency with which the skin is involved in mTORi treatment, mIAS’ limitation in only involving only the oral movable mucosa in lieu of those forms of the oral mucosa that are more like skin (keratinized epithelium) raises another interesting question around mTORi pathobiology. mTORis are typically administered in an oral formulation with a pharmacokinetic profile that is consistent with systemic uptake and distribution. Why, then, are the oral lesions of mIAS not more generalized, rather than distinct?

The primary objective of this manuscript is to present some thoughts, based on what has been already reported, about the possible mechanisms by which mTOR inhibitors induce mucosal damage. The validity of the hypothesis requires investigation. Furthermore, the observation that mIAS risk is inconsistent among patients of similar phenotype and tumor diagnosis suggests the possibility that genomics might contribute to risk determination, or more likely a concert of omics [46].

Oncologic treatment options are advancing quickly as developments in the definition of tumor biology offer drug-targeting opportunities. Likewise, expanding the understanding of host–tumor responses, immunological manipulation, and the potential to modify the tumor environment are actively being exploited in the quest for better cancer treatment options. The uniqueness of the oral cavity with respect to its inclusion of various forms of mucosa, multiple fluids (saliva, crevicular fluid), microflora, immune system (humoral, cellular and innate) and soft and hard tissue types provides a range of potential targets for side effects. There have been no new agents that are free of treatment-related toxicities, and those occurring in the mouth and contiguous tissues have been consistently reported.

Therefore, while the mysteries of mIAS are only partially solved, new oral toxicities have evolved, creating a rich opportunity for study and a need for solutions.

## Figures and Tables

**Figure 1 cancers-16-00068-f001:**
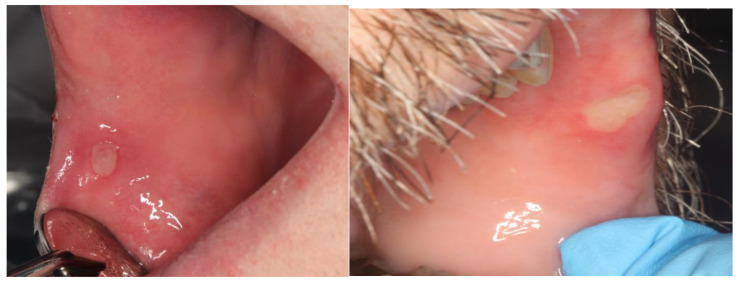
Similarity in clinical presentation of mIAS ((**right**) panel) and recurrent aphthous stomatitis ((**left**) panel).

**Figure 2 cancers-16-00068-f002:**
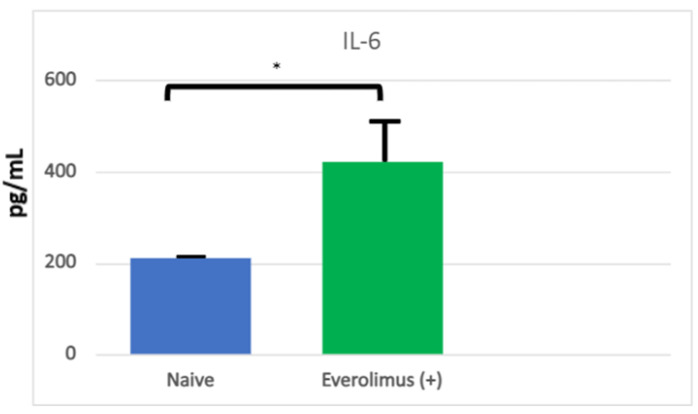
Levels of IL-6 in organotypic human oral mucosal tissue treated with everolimus without or with TFF 1. IL-6 was measured in supernatants from oral mucosal tissue collected 24 h after everolimus stimulation. Supernatants were analyzed for levels of human IL-6 using a MagPix multiplex analyzer. Data are graphed as Mean ± SEM for each group (n = 4) and data were analyzed via One-way ANOVA and Dunnett’s multiple comparison’s test, * *p* < 0.05 compared to Everolimus (+) Control group. From Sonis S, Andreotta PW, Lyng G. On the pathogenesis of mTOR inhibitor-associated stomatitis (mIAS)—studies using an organotypic model of the oral mucosa. Oral Dis 2017; 23:247–52 [19].

**Figure 3 cancers-16-00068-f003:**
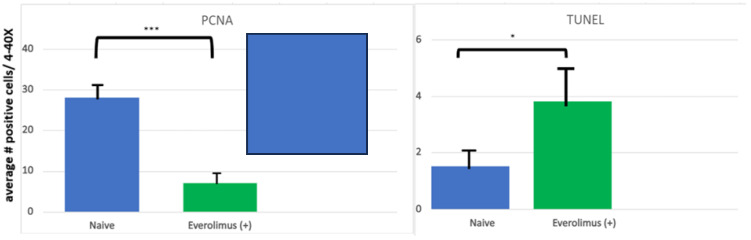
Quantification of PCNA and TUNEL staining in an organotypic model of human oral mucosal tissues. PCNA staining was used to assess cell proliferation. Data are expressed as an average number of PCNA–positive cells across four fields at 20× magnification, averaged per group; data graphed are mean ± SEM per group where n = 3. Data were analyzed via one-way ANOVA and Dunnett’s multiple comparisons. ***: *p* ≤ 0.005 compared to Everolimus (+) group. TUNEL staining was used to assess apoptotic cells and data are expressed as an average number of TUNEL-positive cells across four fields at 40× magnification, averaged per group; data graphed are mean ± SEM per group where n = 3. Data were analyzed via one-way ANOVA and Dunnett’s multiple comparisons. * *p* < 0.05 compared to Everolimus (+) group. From Sonis S, Andreotta PW, Lyng G. On the pathogenesis of mTOR inhibitor-associated stomatitis (mIAS)—studies using an organotypic model of the oral mucosa. Oral Dis 2017; 23:247–52 [19].

**Table 1 cancers-16-00068-t001:** Incidence of mTOR inhibitor-associated stomatitis (mIAS) and related dose interruptions, according to mTOR inhibitor.

	Overall		% Of Adverse Events		Overall	% Of Dose Reductions	Overall	% Of Discontinuations
	All Grades	Grades 3, 4	All Grades	Grades 3, 4				
**All mTOR Inhibitors**	52.9% (1493/2822)	5.4% (153/2822)	73.4% (1493/2033)	20.7% (153/739)	19.2% (176/917)	27.3% (48/176)	11.5% (99/862)	13.1% (13/99)
Temsirolimus	60.8% (819/1347)	5.2% (70/1347)	88.5% (819/925)	16.0% (70/437)	28.1% (140/499)	30% (42/140)	10.2% (40/393)	0 (0/40)
Everolimus	44.3% (568/1281)	5.2% (67/1281)	58.4% (568/972)	27.2% (67/246)	8.2% (28/340)	7.1% (2/28)	12.4% (38/307)	18.4% (7/38)
Ridaforolimus	54.6% (106/194)	8.2% (16/194)	77.9% (106/136)	28.6% (16/56)	10.3% (8/78)	50% (4/8)	13.0% (21/162)	28.6% (6/21)

From Martins F, de Oliveirs MA, Wang Q, Sonis S, Gallottini M, George S, Treister N. A review of oral toxicity associated with mTOR inhibitor therapy in cancer patients. Oral Oncol 2013; 49:292–8. With permission.

## Data Availability

Data are contained within the article.

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
