# Peer review of "A New Hypothesis Describing the Pathogenesis of Oral Mucosal Injury Associated with the Mammalian Target of Rapamycin (mTOR) Inhibitors"

_cancers, 2023, doi:10.3390/cancers16010068_

Round 1
Reviewer 1 Report
Comments and Suggestions for Authors
This is a well-done review to explore the pathogenesis of oral mucosal injury associated with mTOR inhibitors. However, a more systematic review of the literature should be performed to give more solid evidences to support your hypothesis.
Comments on the Quality of English Language
Only one spelling error is found on line 206.
Author Response
Thank you for your kind comments.
Spelling has been reviewed.

Reviewer 2 Report
Comments and Suggestions for Authors
Dear Authors,
Indeed, this paper addresses an interesting idea. Only minor modifications are required before its publicatioDisclosuren.
A. Table 1. At the end of the table title there is given “1”, with no explanation
B. Figure 2 and 3. I recommend using number not the whole title of reference
C. I suggest including meaning of word “rapalog” (i.e. rapamycin and its analogs)
D. Plenty of editorial changes are required (like author’s titles, which should not be placed in the article, or the text beneath Figure 2)
E. I think the Section 5 (disclosure) should be transferred to Conflicts of Interest, also others components of Back Matter are necessary to be completed (according to Instructions for Authors)
F. Finally, why do you want to publish it as a hypothesis, not review? (just out of curiosity)
Best regards and good luck
Author Response
Thank you for your thoughtful comments. Please see the attached point-by-point response.

Reviewer 3 Report
Comments and Suggestions for Authors
This is a very pertinent, well-written and interesting article on the relevance and implications of mTOR inhibitors and their impact on the oral mucosa, not forgetting their mechanistic and systemic effects.
I have only detected a few small details to correct that are not directly related to the content of the article, but rather to flaws in the formatting of the text.
Line 21,22- The sentence is repetitive with the word review used twice
Line 29,30- I think more reference could be added to this sentence
Line 131- Explain that AEs is " adverse events"
Line 141- Remove the "s" from mTORis
Line 142- I think that table 1 should be on the end of the sentence
Line 143,144- I couldn´t see this % on table 1, could you please confirm?
Line 145- Remove ;
Line 158- Format the headings on table 1
Line 161 and 401- correct the year of Rugo publication, is 2016
Line 175- I don´t understand the meaning of granular study
Line 192- Please add some relevant reference to this paragraph
Line 229- Replace MTOR to mTOR
Line 262- Put the reference number on the end of sentence
Line 333 and 337- Use []
Author Response

(The authors gave the same response as above.)

Reviewer 4 Report
Comments and Suggestions for Authors
The topic is new and is of interest to the scientific community.
There are some concerns with the manuscript which need to be addressed by the author before acceptance.
Comments;
Title
1. it will be better if you can expand mTOR Inhibitors in the title
Abstract:
1. aphthous stomatitis (RAS), its recurrent aphthous stomatitis.
2. mIAS’ pathogenesis has been the subject of speculation, and its similar presentation to aphthous stomatitis 20 (RAS) has provoked speculation that it might serve as a surrogate to better understand RAS. (Sentence is difficult to understand, rephrase it so that the readers can understand well)
3. Unexplained unique features of mIAS are discussed in the context of future investigation. (rewrite)
4. Check for typo errors (which mTORis…….)
Body of the text:
1. Check for typo errors and spacing.
2. In the first paragraph of the introduction, many sentences need a citation.
3. Do not use complex English words because the manuscript is to be read by non-English speaking readers.
4. What is TFF1?
5. What is PCNA?
6. What is TUNEL?
7. Unexplained unique features of mIAS are discussed in the context of future investigation. (is this discussed in the manuscript? If yes, please specify)
8. Please concise the conclusion…it’s too lengthy. So be specific in your concluding remark.
Include the rest information in the manuscript.
Author Response

(The authors gave the same response as above.)

Round 2
Reviewer 2 Report
Comments and Suggestions for Authors
Dear Authors,
This is a revised version of your original manuscript. You have corrected it according to the suggestions and it can be accepted without any changes now.
Best regards and good luck